# Mutation Characteristics and Phylogenetic Analysis of Five *Leishmania* Clinical Isolates

**DOI:** 10.3390/ani12030321

**Published:** 2022-01-28

**Authors:** Zhiwan Zheng, Jinlei He, Tao Luo, Jianhui Zhang, Qi Zhou, Shuangshuang Yin, Dali Chen, Jie Luo, Jianping Chen, Jiao Li

**Affiliations:** 1Department of Pathogenic Biology, West China School of Basic Medical Sciences and Forensic Medicine, Sichuan University, Chengdu 610041, China; 18623142171@163.com (Z.Z.); hejinlei818@163.com (J.H.); taoluo@scu.edu.cn (T.L.); zjh1186974551@sina.com (J.Z.); ZQ450710976@outlook.com (Q.Z.); ssyin002@163.com (S.Y.); cdl1978119@sina.com (D.C.); 2West China School of Public Health, Sichuan University, Chengdu 610041, China; ljscuer@163.com; 3Animal Disease Prevention and Food Safety Key Laboratory of Sichuan Province, Chengdu 610065, China

**Keywords:** *Leishmania*, whole-genome resequencing, phylogenetic analysis, mutation characteristics, drug resistance

## Abstract

**Simple Summary:**

Leishmaniasis, a neglected tropical disease, is caused by infection with the *Leishmania* species, threatening millions of people in approximately 100 endemic countries. The emergence of antimony-resistant *Leishmania* strains have brought difficulties to the treatment and elimination of leishmaniasis. This study performed genome-wide resequencing and phylogenetic analysis of five isolates from the *Leishmania donovani* complex, focusing on finding mutations related to antimony resistance and virulence of the newly isolated *Leishmania* strain L_HCZ in 2016. By combining whole-genome sequencing and whole-genome phylogenetic analysis, *Leishmania* isolates L_801, L_9044 and L_Liu were identified as *Leishmania donovani*, and L_HCZ as *Leishmania infantum*. By discovering genome-wide single-nucleotide polymorphisms and structural variations, we identified mutations of drug resistance-related genes in the antimony-resistant *Leishmania* isolate L_HCZ. The new *Leishmania* isolate L_HCZ has strong virulence and strong drug resistance, which should be taken seriously by the relevant health departments and scientific researchers.

**Abstract:**

Leishmaniasis is a neglected tropical disease threatening millions of people worldwide. The emergence of antimony-resistant *Leishmania* strains have brought difficulties to the treatment and elimination of leishmaniasis. This study performed genome sequencing, phylogenetic analysis and mutation analysis of five *Leishmania* clinical isolates, especially the *Leishmania* strain L_HCZ isolated in 2016, which shows strong virulence and antimony resistance. By phylogenetic analysis, four isolates (L_DD8, L_801, L_Liu and L_9044) were identified as *Leishmania donovani*, the isolate L_HCZ was identified as *Leishmania infantum* and the isolate L_DD8 as a standard strain of *L**.donovani*. Genome-wide mutation analysis was applied to identify mutations related to the drug resistance and virulence of the newly isolated L_HCZ. Compared with the other four *Leishmania* isolates, L_HCZ had the most mutations in genes associated with antimony resistance, including the ABC transporter, ascorbate-dependent peroxidase, gamma–glutamylcysteine synthetase, glucose-6-phosphate 1-dehydrogenase, ATP-binding cassette protein subfamily A and multi-drug resistance protein-like genes. Among the genes associated with virulence, L_HCZ had the most mutations in cysteine peptidase A, cysteine peptidase B, cysteine peptidase C, heat-shock protein 70, gp63, acid phosphatase, kinesin k39, kinesin, phosphoglycan beta 1, amastin-like surface protein and amastin-like proteins. The mutations in L_HCZ might possibly contribute to its antimony resistance and strong virulence in clinical patients. Whole-genome resequencing has exhibited broad application prospects and may be put into clinical use in the future for parasite identifying and epidemiological investigations.

## 1. Introduction

Visceral leishmaniasis (VL) is a neglected zoonotic parasitic disease and is caused by *Leishmania* parasites, which are transmitted by insects known as sandflies. The main clinical symptoms of VL are long-term irregular fever, anemia, weight loss and hepatosplenomegaly [1]. Failure to treat the infection in time leads to the death of patients. The *Leishmania* spp. that causes VL is mainly derived from the *Leishmania donovani* complex (including *Leishmania donovani* and *Leishmania infantum*) [2]. At present, VL is endemic in more than 60 countries worldwide and an estimated 50,000 to 90,000 new cases of VL occur worldwide annually [3]. Only a few anti-*Leishmania* drugs are available, and their effectiveness is severely limited by toxicity, cost and drug resistance [4]. Some resistant *Leishmania* strains have appeared in clinics, especially with resistance to antimony. Understanding the *Leishmania* genomes, their natural variation and genetic relationships is essential to support and strengthen public health monitoring and intervention strategies.

*Leishmania* is a protist belonging to the family trypanosomidae [5,6]. Currently, about 53 *Leishmania* species are known, and at least 20 species are pathogenic for human beings [7]. The classification of *Leishmania* was initially based on eco-biological criteria such as vector, geographical distribution, tropism, antigenic properties and clinical manifestations [8]. With the development of modern technology, the classification of *Leishmania* began to be based on combined molecular data. The currently accepted classification of *Leishmania* species is divided into two major phylogenetic lineages: *Euleishmania* and *Paraleishmania* [7]. The *Euleishmania* section is composed of at least four subgenera: *Leishmani*a, *Viannia*, *Sauroleishmania* and *Mundinia* [9,10,11]. The subgenus *Leishmania* has four main species complexes: *L. donovani*, *L. major*, *L. mexicana* and *L. tropica* [9,11]. A reliable taxonomy of *Leishmania* species will represent a keystone for biological and epidemiological research programs and can help guide clinical treatment and prevention [7]. However, there is still no universal agreement regarding the classification of *Leishmania* at the specific or subspecific level. The advent of molecular sequence data provided a large extent of additional information for phylogenetic analyses [8]. Molecular markers such as the internal transcribed spacer (ITS) 1 and 2 of the ribosomal DNA array, the cytochrome B gene (*cyt*B) and the heat-shock protein 70 gene (*hsp*70) were used for the phylogenetic analysis of *Leishmania*, but some evolutionary relationships are still not well resolved.

The genome size of *Leishmania* varied among different subspecies, *L. amazonensis* genome is 29 Mb in size, while *L. infantum*, *L. major* and *L. braziliensis* genomes are 32 Mb in size [10,12,13]. These genomes are organised into a variable number of chromosomes, 34 in *L. amazonensis* and *L. mexicana*, 35 in *L. brasiliensis* and 36 in *L. major*, *L. donovani* and *L. infantum* [12,14]. The *Leishmania* genome has more than 8300 coding genes [15]. More than 99% of the genes between *L. infantum*, *L. major* and *L. braziliensis* are conserved, and only a few species-specific genes were found [13]. The lack of species-specific genes suggests that differences in disease predisposition and drug sensitivity between species may be related to differences in gene expression and gene doses of common core genes [16,17]. With the rapid development of sequencing technology, next-generation sequencing technology has enabled high-throughput and genome-scale screening of eukaryotic pathogens and has played an important role in identifying drug targets and elucidating drug resistance mechanisms [18]. In recent years, the use of whole-genome data to construct phylogenetic trees has become a new direction of phylogenetic research. Whole-genome data can effectively eliminate the influence of factors such as horizontal gene transfer and differences in gene evolution rate between groups on the phylogenetic tree [19]. Moreover, genome-wide single-nucleotide polymorphism (SNP) typing represents a powerful alternative approach for differentiating parasite strains [20]. In this study, five *Leishmania* clinical isolates were collected, especially the newly isolated strain MHOM/CN/2016/SCHCZ (L_HCZ) [21]. The L_HCZ strain showed resistance to antimony and strong virulence in a clinical patient, resulting in severe clinical symptoms. We performed whole-genome resequencing and phylogenetic analysis on these *Leishmania* isolates and looked for mutations in the genome that are potentially related to antimony resistance and virulence, especially in the L_HCZ isolate.

## 2. Materials and Methods

### 2.1. Parasites

Five *Leishmania* clinical isolates were used in this study (Table 1). All five *Leishmania* isolates belong to the *Leishmania donovani* complex. The isolates L_801, L_9044, L_DD8 and L_Liu were presented by the National Institute of Parasitic Diseases (IPD), Chinese Center for Disease Control and Prevention (China, CDC). L_HCZ is a new *Leishmania* strain isolated from the bone marrow of a patient with kala-azar by our experimental team.

### 2.2. Leishmania Culture and Acquisition of Genomic DNA

The promastigotes of the five *Leishmania* isolates were preserved in Novy–MacNeal–Nicolle (NNN) medium. The parasites, preserved in NNN medium, were transferred to M199 medium (Sigma-Aldrich, St. Louis, MO, USA) supplemented with 10% fetal bovine serum (FBS; Gibco, Franklin, TN, USA) and antibiotics (100 U/mL penicillin and 100 μg/mL streptomycin; Hyclone, Logan, UT, USA) at pH 7.4 and 28 °C for extended culture. After cultivating the parasites for 5–7 days to reach the logarithmic growth phase, the medium was added logarithmically. The morphology of *Leishmania* promastigotes was observed by Wright’s staining. When the total amount of M199 medium reached 80 mL or more, the parasites were collected by centrifuge at 800× *g* for 10 min to extract genomic DNA using TIANamp Genomic DNA Kit (Tiangen, Beijing, China). The concentration of DNA was detected by Qubit Fluorometer, and the integrity of DNA was detected by agarose gel electrophoresis. The qualified DNA was sent to The Beijing Genomics Institute for sequencing.

### 2.3. Whole-Genome Resequencing and Analysis

#### 2.3.1. Reference Genome

The reference genome (GCF_000227135.1) of the strain *Leishmania donovani* BPK282A1 was used. The genome contains sequences of 36 chromosomes and is 32,444,968 bp in length in total. The GC content of the genome is 59.40%, and the gap rate of the genome is 3.676%. The N50 length of the genome is 1.024 MB and the N90 of the genome is 558.17 KB.

#### 2.3.2. Variant Detection and Annotation

The main steps of GATK best practices for variation detection were as follows: clean reads obtained by sequencing were compared with the reference genome using BWA software [22]. Picard was used to mark the repeated alignment results (Parameter: MarkDuplicates), sort the alignment results (Parameter: SortSam SORT_ORDER = “coordinate”) and correct system errors in base quality scores (Parameter: BaseRecalibrator–use–original–qualities). GATK4 [23] was used to detect single nucleotide polymorphisms (SNPs) and insertion/deletions (InDels) (Parameter: HaplotypeCaller–ERC GVCF) to obtain GVCF result files and used to combine all samples of GVCF for joint detection of SNPs + InDels (Parameter: GenotypeGVCFs) to obtain genotype files for all samples. Parameters for SNP filtering: ExcessHet > 54.69, QD < 2.0, MQ < 40.0, FS > 60.0, SOR > 3.0, MQRankSum < −12.5, ReadPosRankSum < −8.0. INDEL filtering: ExcessHet > 54.69, QD < 2.0, FS > 200.0, SOR > 10.0, MQRankSum < −12.5, ReadPosRankSum < −8.0. Filtered SNPs and InDels were annotated using Annovar [24] (http://annovar.openbioinformatics.org/en/latest/, accessed on 9 August 2020). CNVnator [25] (v.0.3, https://github.com/abyzovlab/CNVnator, accessed on 9 August 2020) to detect copy number variation (CNV) and BreakDancer [26] (v.1.1, https://github.com/genome/breakdancer, accessed on 9 August 2020) to detect structural variation (SV).

### 2.4. Genome Assembly and Phylogenetic Analysis

We downloaded 28 genome sequences of *Leishmania* strains from NCBI (Table 2). These complete genome sequences, together with the complete genome sequences of the five *Leishmania* isolates we obtained, were used for phylogenetic analysis. The genomes of five *Leishmania* isolates were sequenced by Illumina Hiseq 2000 in paired-end mode. The fastq reads from current study were firstly trimmed by fastp [27] (v0.20.0, https://github.com/OpenGene/fastp, accessed on 21 October 2021) with a quality threshold of 20, a length threshold of 50 bp and a front trimming of 5 bp in both ends. The trimmed reads were subjected to assembly by Shovill (v1.0.0, https://github.com/tseemann/shovill, accessed on 21 October 2021) with default parameters. Contigs with low coverage of less than 3 and lengths less than 1000 bp were filtered. FastANI [28] (v1.2, https://github.com/ParBLiSS/FastANI, accessed on 21 October 2021) was used to calculate pair-wise genomic similarities between all 33 genomes and a similarity matrix was generated. MultiDendrograms [29] (v5.1, https://github.com/sergio-gomez/MultiDendrograms, accessed on 21 October 2021) was used to construct the phylogeny of the strains based on the similarity matrix with the UPGMA algorithm.

The complete or draft genomes of the 33 strains were mapped to the genome of the strain Friedlin (GCA_000002725.2) by Minimap2 [30] (v2.1, https://lh3.github.io/minimap2/, accessed on 21 October 2021) and core regions with a length ≥ 1000 bp for each were determined by Samtools [31] (v1.10, http://samtools.github.io, accessed on 21 October 2021) and Bedtools [32] (v2.30.0, https://bedtools.readthedocs.io, accessed on 21 October 2021). The sequences from the core regions were extracted by Maf parase [33] (v1.42, https://github.com/CshlSiepelLab/phast, accessed on 21 October 2021) from the MAF file generated by minimap2. The sequences from each core region were aligned using Mafft [34] (v7.487, https://mafft.cbrc.jp/alignment/software/, accessed on 21 October 2021) and then trimmed by Trimal [35] (v1.4.rev15, http://trimal.cgenomics.org, accessed on 21 October 2021). The alignments of all the core regions were concatenated into a single core-genome alignment, which was then used for phylogenetic analysis by iqtree [36] (v1.6.12, http://www.iqtree.org/, accessed on 21 October 2021) using the TVM + F + G4 model as determined automatically and by 1000 bootstrap replicates.

### 2.5. Mutation Analysis in Virulence and Resistance Associated Genes

As L_HCZ showed obvious resistance to sodium stibogluconate and strong pathogenicity in the clinical patient, we focused on the genetic mutation of L_HCZ and compared it with the other four *Leishmania* isolates. Nine genes related to antimony resistance were selected for this study, which were ABC transporter, aquaglyceroporin, ascorbate-dependent peroxidase, dihydrofolate reductase–thymidylate synthase, gamma–glutamylcysteine synthetase, glucose-6-phosphate 1-dehydrogenase, mitogen-activated protein kinase, sodium stibogluconate resistance protein and trypanothione reductase [37,38]. Nine important genes related to the virulence of *Leishmania* were also selected for mutation analysis, which were cysteine peptidase A, cysteine peptidase B, cysteine peptidase C, heat-shock protein 70, cyclophilin-40, gp63, surface protein amastin, acid phosphatase and kinesin k39 [39]. Exon mutations in nine drug resistance genes and nine virulence genes of five *Leishmania* isolates were counted, mainly including synonymous mutations, nonsynonymous mutations, frameshift insertion, frameshift deletion, non-frameshift insertion, non-frameshift deletion, stopgain and stoploss. Moreover, compared with the reference genome, we found a great number of exon mutations in some genes of L_HCZ. We counted genes with more than 50 exon mutations in L_HCZ and compared them with the other four *Leishmania* isolates.

## 3. Results

### 3.1. Resequencing Analysis

#### 3.1.1. Mapping Results with Reference Genome

The sequencing reads of five samples were aligned to the reference genome. The mapping rate of L_Liu was the highest (93.93%), followed by isolates L_801, L_DD8 and L_DD8, and the isolate L_HCZ had the lowest map rate (90.01%). The genomic coverages based on 1×, 4×, 10× depths for the samples were all greater than 90%, suggesting that the five isolates had high similarity with the reference genome (Table 3).

#### 3.1.2. Detection Results of SNPs and InDels

Some differences in the number of SNPs and InDels among the five *Leishmania* isolates were found (Figure 1). The number of SNPs, insertions, deletions and InDels of L_801 and L_DD8 were basically the same and small, suggesting that they were close to the reference genome. The total Het/Hom ratio and SNP Het/Hom ratio of L_801 and L_DD8 were also basically consistent and high, reflecting that their heterozygous variations were more than homozygous variations. Moreover, the SNP Transitions/Transversions ratios of L_801 and L_DD8 were also basically the same, but they were lower than those of other *Leishmania* isolates. The number of SNPs, insertions, deletions, total Het/Hom ratio, SNP Het/Hom ratio and SNP Transitions/Transversions of L_9044, L_Liu and L_HCZ was approximately the same, and their homozygous variations were more than heterozygous variations, which was different from the results of L_801 and L_DD8. The number of InDels of L_Liu and L_9044 was higher than that of other *Leishmania* isolates.

The distribution of SNPs and InDels of the five *Leishmania* isolates is shown in Figure 2. Similarly, the distribution of SNPs and InDels of L_801 and L_DD8 was different from that of the other three isolates. The SNPs and InDels of L_801 and L_DD8 were mainly distributed upstream and downstream of genes, followed by the exonic region. However, the SNPs and InDels of L_9044, L_Liu and L_HCZ were most distributed in the exonic region, followed by upstream and downstream of genes, and their distribution in the upstream and downstream of genes was roughly the same. Only a very small number of SNPs and InDels in the five *Leishmania* isolates was distributed in the exonic region of ncRNA and the 5′ UTR region. It is worth noting that none of the five *Leishmania* SNPs and InDels were distributed in intronic, ncRNA_intronic, 3′ UTR and splicing regions.

The number of different types of mutations caused by SNPs and InDels in the coding region of the five *Leishmania* isolates is shown in Figure 3. L_801 and L_DD8 were slightly less synonymous than nonsynonymous, and the number of frameshifts was significantly more than that of non-frameshifts, and the number of stopgains was significantly higher than that of stoplosses. Among the mutations caused by the SNPs and InDels of L_9044, L_Liu and L_HCZ, the number of synonymous ones was basically equivalent to that of nonsynonymous ones, and the number of frameshifts was less than that of non-frameshifts, and the number of stopgains was more than that of stoplosses.

#### 3.1.3. Detection Results of CNV and SV

As shown in Figure 4A, in CNV detection results, the number of duplications of all *Leishmania* isolates was greater than that of deletion, but their proportions of duplication and deletion were different. Similarly to the results of SNPs and InDels, the CNV results of L_801 and L_DD8 were also different from those of the other three *Leishmania*. The proportion of duplication of L_801 and L_DD8 was 62.8~67.2%, while that of L_9044, L_Liu and L_HCZ was 72.9~77.2%.

The SV results of L_801, L_DD8, L_9044 and L_Liu were similar, and their proportions of deletion (about 35%), insertion (about 3%), inversion (about 10%), intrachromosomal translocation (about 32%) and interchromosomal translocation (about 20%) were basically the same (Figure 4B). It is worth noting that the results of L_HCZ were different from those of the other four isolates, and its proportions of deletion, insertion, inversion, intrachromosomal translocation and interchromosomal translocation were 30.4%, 0.5%, 10.4%, 30.1% and 28.6%, respectively. Compared with the other four isolates, the proportions of deletion and insertion of L_HCZ decreased, and the proportion of interchromosomal translocation increased.

### 3.2. Whole-Gene Phylogenetic Analysis

The whole-genome sequences of 28 *Leishmania* strains were downloaded from NCBI and analyzed together with the whole-genome sequences of the five *Leishmania* clinical isolates. The pairwise comparison matrix of the whole genomic nucleic acids of the 33 *Leishmania* strains is shown in Figure 5. A core-genome alignment with a length of 7.3 Mbp was generated for phylogenetic analysis (Figure 6) based on the maximum-likelihood algorithm. These *Leishmania* strains were divided into four groups: *L. donovani* complex, *L. major* complex, *L. tropica* complex and *L. mexcicana* complex. The results of the phylogenetic tree show that L_801, L_DD8 and three *L. donovani* strains were clustered into a small branch, proving that they were closely related. The relationship between L_DD8 and L_801 also showed a high correlation on the matrix diagram. L_DD8 is the internationally common standard strain of *L. donovani*. Therefore, it could be determined that L_801 is *L. donovani*. L_9044 and L_Liu were clustered together in the tree, suggesting that their evolutionary relationship is close, but they were slightly distant from the rest of the *L. donovani* strains. L_9044 and L_Liu were temporarily incorporated into the *Leishmania donovani* complex. Due to the close evolutionary relationship between L_801 and L_DD8, the SNPs, InDels and CNV data of L_801 and L_DD8 were basically consistent and could be explained. Similarly, the consistent SNPs, InDels and CNV results of L_9044 and L_Liu may also be caused by their close evolutionary relationship. It should be noted that L_HCZ was clustered with two canine *L. infantum* strains and one human *L. infantum* strain in the tree. Consequently, the L_HCZ isolate could be identified as *L. infantum*. However, because of the close genetic relationship between *L. donovani* and *L. infantum* [40], the genome-wide phylogenetic tree in this study could not completely separate *L. donovani* and *L. infantum*.

### 3.3. Mutation Analysis of L_HCZ

#### 3.3.1. Exon Mutations in Five *Leishmania* Isolates

Compared with the reference genome, L_DD8 had 883 mutations in exons, 773 mutations were aneuploid, and the proportion of aneuploidy was 87.54%. L_801 had 1022 mutations in exons, 918 mutations were aneuploid, and the proportion of aneuploidy was 89.82%. L_9044 had 37,796 mutations in exons, 1648 mutations were aneuploid, and the proportion of aneuploidy was 4.36%. L_Liu had 38,078 mutations in exons, 2054 mutations were aneuploid, and the proportion of aneuploidy was 5.39%. L_HCZ had 49,352 mutations in exons, 1322 mutations were aneuploid, and the proportion of aneuploidy was 2.68%. We found that although L_HCZ, L_Liu and L_9044 had more exon mutations, most of them were euploid. However, although L_801 and L_DD8 had fewer exon mutations, most of them were aneuploid (Table 4).

#### 3.3.2. Mutation Analysis in Nine Genes Associated with Antimony Resistance

The number of mutations in nine antimony resistance-related genes of five *Leishmania* isolates is shown in Figure 7. Most of the genes had only nonsynonymous and synonymous mutations, and only dihydrofolate reductase–thymidylate synthase (frameshift deletion), sodium stibogluconate resistance protein (frameshift deletion, frameshift insertion, non-frameshift deletion, non-frameshift insertion and stopgain) and trypanothione reductase (frameshift deletion and frameshift insertion) had other types of mutations. Among the nine genes, only sodium stibogluconate resistance protein and aquaglyceroporin had aneuploidy mutations. L_DD8 and L_801 only had mutations in the sodium stibogluconate resistance protein gene, and no mutations were found in the other eight genes. The nine genes in L_9044, L_Liu and L_HCZ all had varying degrees of mutation. The mutation types, numbers and sites of the nine genes in L_9044 and L_Liu were mostly the same, which was consistent with the results of previous resequencing and phylogenetic analysis. Except for dihydrofolate reductase–thymidylate synthase, mitogen-activated protein kinase and sodium stibogluconate resistance protein, the other genes had the most mutations in L_HCZ.

#### 3.3.3. Mutation Analysis in Nine Genes Associated with Virulence

The number of mutations in nine virulence genes of five *Leishmania* isolates is shown in Figure 8. Only gp63 (nonframeshift deletion, nonframeshift insertion and stoploss), acid phosphatase (nonframeshift deletion) and kinesin k39 (frameshift insertion) genes had other types of mutations, while other genes had only nonsynonymous and synonymous mutations. Among the nine genes, only cysteine peptidase B, gp63, surface protein amastin and kinesin k39 had aneuploidy mutations. L_DD8 and L_801 only had synonymous mutations in kinesin k39 gene, and the numbers and gene sites of synonymous mutations in the two isolates were identical. Similar to the results of antimony resistance-related genes, the mutation types, numbers and sites of the nine virulence genes in L_9044 and L_Liu were mostly the same. Except for cyclophilin-40 and surface protein amastin, the other genes had the most mutations in L_HCZ.

#### 3.3.4. Mutation Analysis of 18 Genes

We counted genes with more than 50 mutations in the L_HCZ isolate and finally obtained 18 genes. Genes involved in cell signaling and various biological functions: protein kinase [41], ubiquitin hydrolase [41], serine/threonine-protein kinase [42], phosphatidylinositol 3-kinase-like protein [43] and receptor-type adenylate cyclase a [44]. Genes related to the virulence of *Leishmania*: kinesin [45], calpain-like cysteine peptidase [46], phosphoglycan beta 1 [47], amastin-like surface protein [48] and amastin-like protein [48]. A gene related to cilia and flagella movement: dynein heavy chain [49]. A gene involved in protein synthesis and processing: RNA binding protein [50]. A gene related to cell metabolism: N-acyl-L-amino acid amidohydrolase [51]. Genes associated with drug resistance in *Leishmania*: ATP-binding cassette protein subfamily A [52], folate/biopterin transporter [53] and protein-like multidrug resistant [52]. Unknown functional gene: uncharacterized protein and kinetoplast-associated protein-like protein. The number of mutations in the 18 genes of five *Leishmania* is shown in Table 5. Most of these genes had the highest number of mutations in the L_HCZ isolate.

## 4. Discussion

Recently, the development of whole-genome sequencing and computational analysis has improved the resolution and accuracy of in-depth exploration and comparison of *Leishmania* genomes [54]. Although the various data of L_HCZ sequencing are close to the data of L_9044 and L_Liu, we still found some special features in the results of L_HCZ sequencing. Compared with the other four *Leishmania* isolates, L_HCZ had the largest number of SNPs and deletions, and had the largest number of variations in upstream, downstream, the exonic region of ncRNA and the 5′ UTR region. Furthermore, in coding regions, the number of SNV, insertion, deletion, synonymous, nonsynonymous, frameshift and non-frameshift mutations was also the highest in L_HCZ. It is interesting that L_HCZ had the lowest number of CNV and SV, especially the number of CNV duplication. The CNV duplication number of L_DD8 and L_801 was almost twice that of L_HCZ. Gene duplication would represent an important means of innovation by expanding the genomic information, and normal copies of the gene would compensate deleterious mutations, whereas successful mutations could be expanded [55]. In trypanosomatids, gene expression is regulated primarily at the post-transcriptional level rather than at the initial level and allows the up-regulation of expression through copy number amplification [55,56]. Therefore, whether these mutations have functional consequences needs to be further explored at the proteomic and metabonomic levels.

The results of the phylogenetic tree corresponded to the SNP/InDel results of five *Leishmania* isolates. In the phylogenetic tree, L_DD8 and L_801 were clustered together, while the data of the two isolates were similar in the SNP/InDel results. The same was true for the phylogenetic and SNP/InDel results of L_9044 and L_Liu. Genome-wide SNP variation provided new evolutionary insights into the ongoing diversification of the phenotypically variable set of *Leishmania* strains [57]. Based on genome-wide SNP/InDel variation analysis and phylogenetic analysis, it could be basically determined that the controversial L_801 in Genebank could be identified as *L. donovani* and had a close genetic relationship with L_DD8. L_Liu was isolated from a patient with cutaneous leishmaniasis in Karamay, Xinjiang, China [58]. An earlier study of cutaneous leishmaniasis in Karamay, Xinjiang found that the pathogen was *L. infantum* [59]. However, in our study, the results of both the phylogenetic tree and SNP/InDel show a close genetic relationship between L_9044 and L_Liu, which is basically consistent with the results of Zhang et al. [58]. In the previous studies, L_9044 and L_Liu were classified as *L. donovani* [58,60]. Therefore, in combination with our genome-wide phylogenetic tree, L_9044 and L_Liu should be identified as *L. donovani*. Moreover, L_HCZ was grouped with three *L. infantum* isolates in the phylogenetic tree, which basically confirmed L_HCZ as *L. infantum*. Whole-genome resequencing is economical, fast and convenient. It can be used for the identification of *Leishmania* isolates and searches for drug resistance targets. It has broad application prospects and may be put into clinical use with phylogenetic analysis in the future for identifying pathogens and epidemiological investigations [57,61].

SNPs in drug targets or transporters can cause drug resistance without changing gene expression, so single-nucleotide mutations constitute a strategy to avoid drug pressure [62]. Single-nucleotide mutations can lead to drug resistance by interfering with drug-target interactions or by changing drugs’ permeability to cells [37]. Another genomic mechanism for drug resistance in *Leishmania* is to regulate the expression of drug target, drug transporter or other drug resistance determinants by changing gene dose through copy number variations (aneuploidy, gene amplification or gene deletion) [62,63]. In this study, compared with the other four *Leishmania* isolates, the number of mutations in virulence genes and drug resistance-related genes of L_HCZ was larger. Except for L_HCZ, the other four *Leishmania* isolates were isolated from the 1980s and 1990s. As a newly isolated *Leishmania* strain, the mutations in virulence genes and drug resistance-related genes of L_HCZ might be caused by the variation accumulation during its evolution [64]. The mutations in L_HCZ might be one of the reasons for its antimony resistance and strong virulence in the clinical patient. Furthermore, in this study, *L. infantum* (L_HCZ) had more mutations in virulence genes than those of *L. donovani* (L_DD8, L_801, L_9044 and L_Liu), which might be one of the reasons why *L. infantum* is more virulent than *L. donovani*. In this study, we investigated the mutation characteristics of five *Leishmania* clinical isolates based on the results of whole-genome resequencing, and analyzed the phylogenetic relationship of these isolates by constructing a phylogenetic tree with the published whole-genome sequences of 28 *Leishmania* isolates. Genome-wide phylogenetic analysis would be helpful to improve the molecular identification of *Leishmania* species, and genome-wide mutation analysis revealed possible mechanisms of pathogenicity and drug resistance in different *Leishmania* isolates. The datasets of the five *Leishmania* clinical isolates will provide a molecular basis for future research on the pathogenic mechanism, drug resistance mechanism and vaccine development of *Leishmania*. In our previous study, we focused on verifying the virulence and antimony resistance of the *Leishmania* clinical isolate (L_HCZ) at molecular levels and proved that L_HCZ has high-level pathogenicity and evident antimony resistance in vitro, using an integrative approach of genome sequencing, proteomic profiling and phenotypic analysis to find the target spot of virulence and antimony resistance of L_HCZ [65]. According to these two papers, some interesting results attracted our attention. In this study, we found the genes of protein kinase, kinesin, amastin-like surface protein, N-acyl-L-amino acid amidohydrolase protein and sodium stibogluconate resistance protein in L_HCZ had more mutations, and in our previous study we found the expression of these genes at the protein level was up-regulated [65]. Similarly, the mutation of mitogen-activated protein kinase in L_HCZ was less than that in L_9044 and L_Liu, and the protein expressed by this gene was down-regulated. Therefore, these results suggest that mutations in these genes affected the expression levels of the corresponding proteins. The gene mutations of protein kinase, kinesin, amastin-like surface protein, N-acyl-L-amino acid amidohydrolase protein and sodium stibogluconate resistance protein in L_HCZ up-regulated the expression of the corresponding proteins and might play roles in the pathogenesis and drug resistance of L_HCZ. Virulence genes, drug resistance-related genes and genes with a high incidence of mutations selected from these *Leishmania* isolates may become targets for vaccines or drugs, as well as the molecular markers for screening of *Leishmania* drug-resistant strains and different virulence strains [38].

## 5. Conclusions

This study performed genome-wide resequencing and phylogenetic analysis of five isolates from the *Leishmania donovani* complex, focusing on finding mutations related to antimony resistance and virulence of the newly isolated *Leishmania* strain L_HCZ in 2016. By combining whole-genome sequencing and whole-genome phylogenetic analysis, *Leishmania* isolates L_801, L_9044 and L_Liu were identified as *L. donovani*, and L_HCZ was *L. infantum*. By discovering genome-wide single-nucleotide polymorphisms and structural variations, we identified mutations of drug resistance-related genes in the antimony-resistant *Leishmania* isolate L_HCZ. The new *Leishmania* isolate L_HCZ has strong virulence and strong drug resistance, which should be taken seriously by relevant health departments and scientific researchers.

## Figures and Tables

**Figure 1 animals-12-00321-f001:**
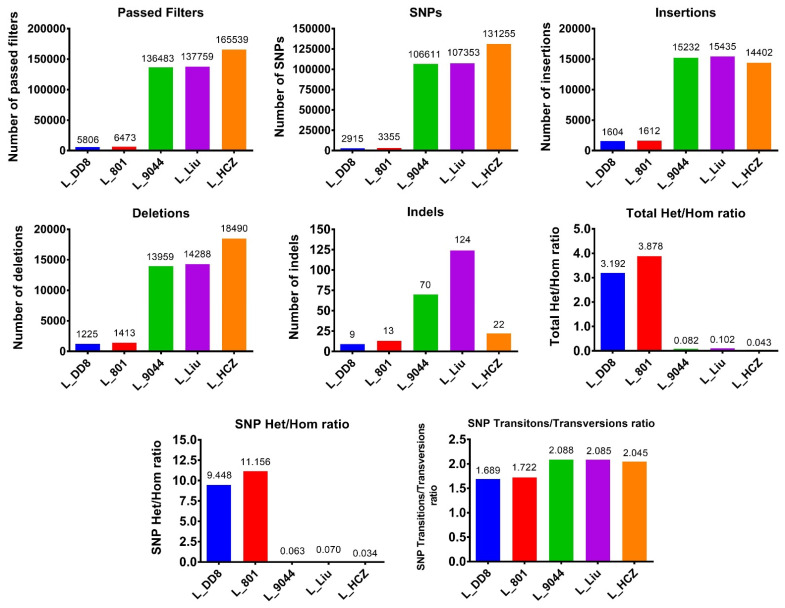
SNPs/InDels detection results. Passed Filters: variations verified by the GATK quality control process. InDels: Insertions/Deletions. Total Het/Hom ratio: the ratio of total heterozygous variation to homozygous variation. SNP Het/Hom ratio: the ratio of heterozygous SNP to homozygous SNP. SNP Transitions/Transversions: the ratio of SNP transitions to transversions.

**Figure 2 animals-12-00321-f002:**
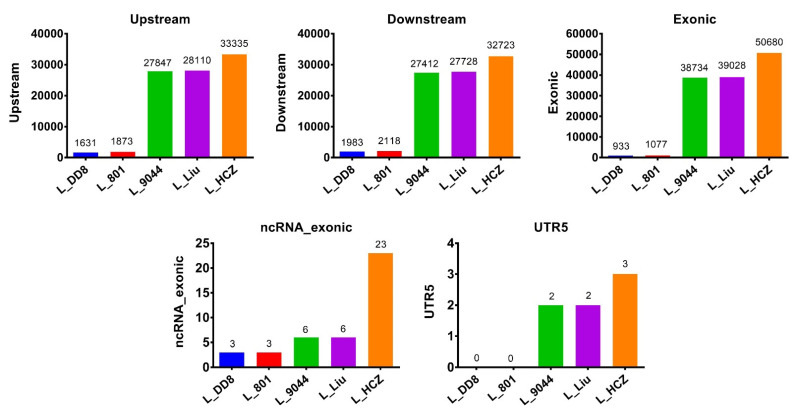
SNPs/InDels distributions. Upstream: variations located upstream of genes. Downstream: variations located downstream of genes. Exonic: variations located at the exonic region of genes. ncRNA_exonic: variations located at the exonic region of ncRNA. UTR5: variations located at the 5′ end of genes.

**Figure 3 animals-12-00321-f003:**
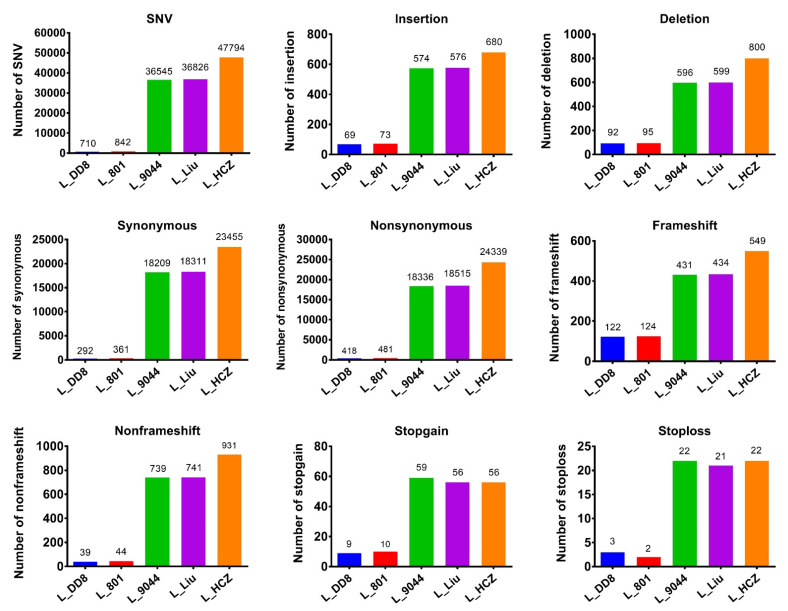
SNPs/InDels in the coding region. SNV: single-nucleotide variations. Stopgain: variations leading to early termination of codon in the coding region. Stoploss: variations leading to loss of normal termination codon in the coding region.

**Figure 4 animals-12-00321-f004:**
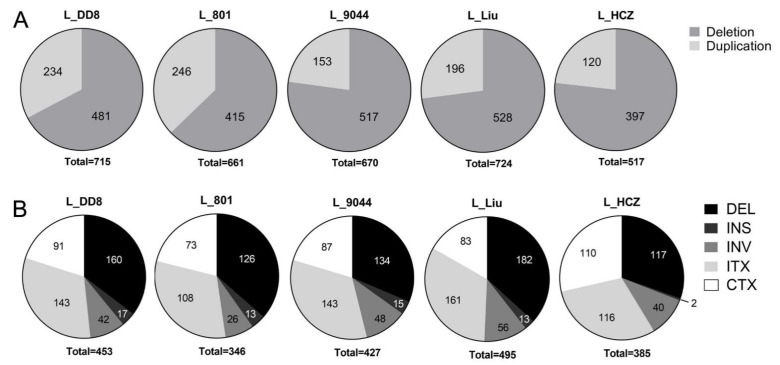
CNV and SV results. (**A**) Copy number variations of the five *Leishmania* isolates. (**B**) Structural variations of the five *Leishmania* isolates. DEL: Deletion. INS: Insertion. INV: Inversion. ITX: Intrachromosomal translocation. CTX: Interchromosomal translocation.

**Figure 5 animals-12-00321-f005:**
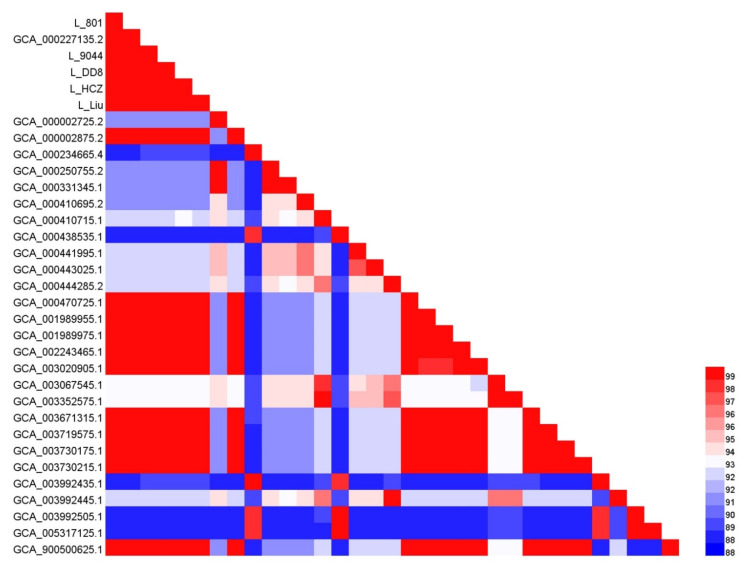
Whole-genome average nucleotide identity matrix of the 33 *Leishmania* strains. The heatmap was generated by the software HemI (Heatmap Illustrator, version 1.0.3.7) and shows the similarity value of genomic nucleic acids in pairwise comparison. Red represents the high similarity value of genomic nucleic acids, and blue represents the low similarity value of genomic nucleic acids.

**Figure 6 animals-12-00321-f006:**
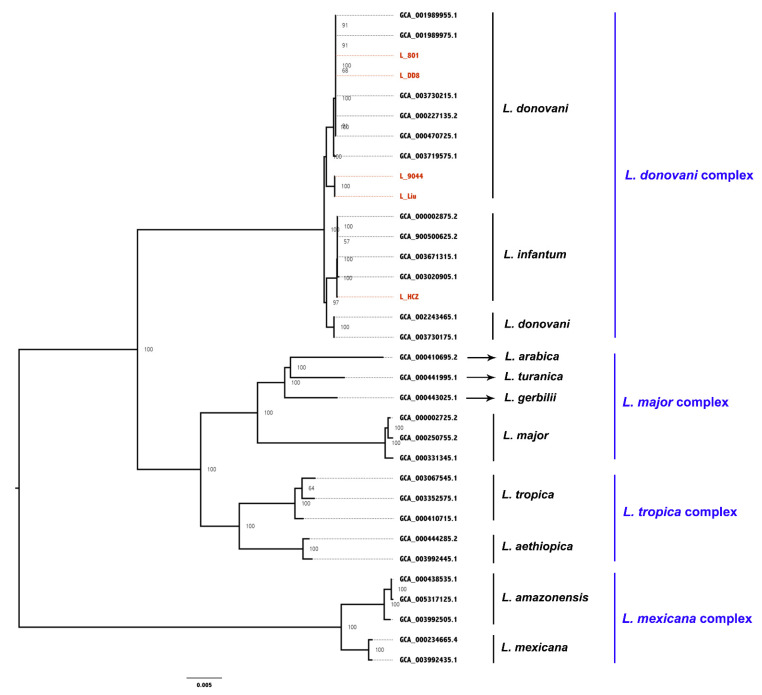
The core-genome based maximum-likelihood phylogeny of the 33 *Leishmania* strains. Node label, bootstrap values. Tip label, GenBank accession or sample name of the 33 strains. The five strains sequenced in the current study are highlighted in red, *L*.: *Leishmaina*.

**Figure 7 animals-12-00321-f007:**
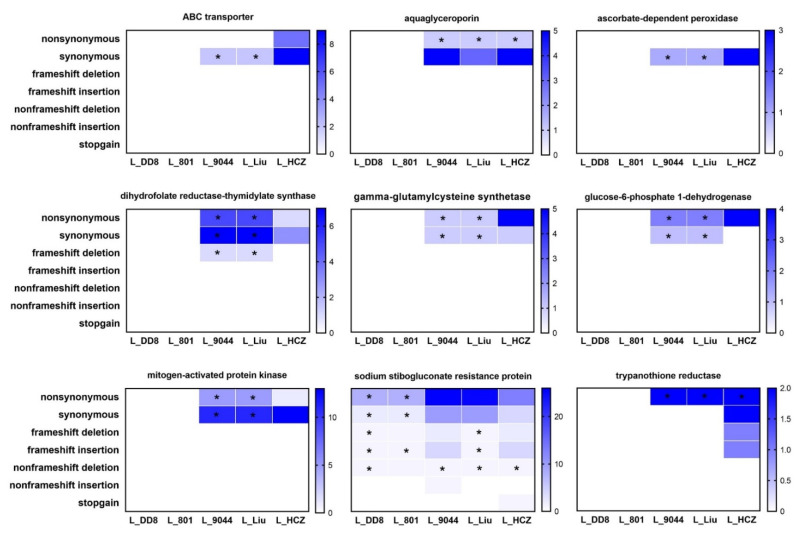
Mutation analysis of antimony resistance-related genes in five *Leishmania* isolates. * Means that in this type of mutation, the gene sites, numbers and bases of mutations in different *Leishmania* isolates were completely consistent.

**Figure 8 animals-12-00321-f008:**
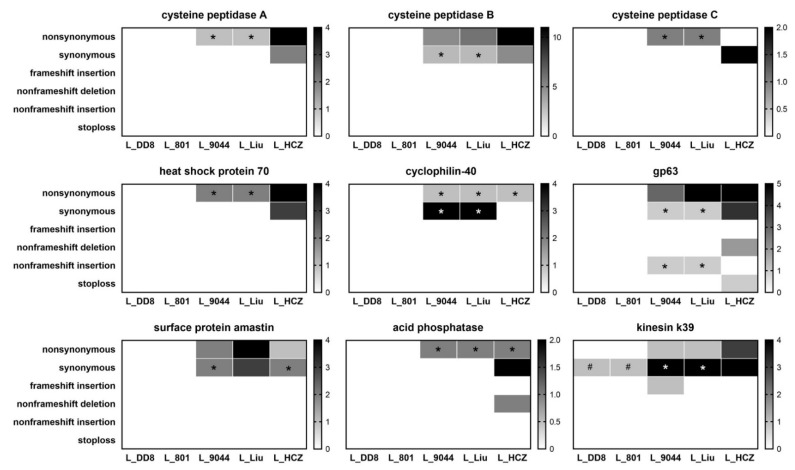
Mutation analysis of virulence genes in five *Leishmania* isolates. * Means that in this type of mutation, the gene sites, numbers and bases of mutations in L_9044, L_Liu and L_HCZ were completely consistent. # Means that in this type of mutation, the gene sites, numbers and bases of mutations in L_DD8 and L_801 were completely consistent.

**Table 1 animals-12-00321-t001:** Information of the five *Leishmania* clinical isolates in this study.

*Leishmania* Species	Isolate	Country	Host	Abbreviation in This Study
*Leishmania donovani*	MHOM/IN/80/DD8	India	Homo	L_DD8
*Leishmania donovani* ^a^	MHOM/CN/80/801	Kashgar, Xinjiang, China	Homo	L_801
*Leishmania donovani*	MHOM/CN/90/9044	Shandong, China	Homo	L_9044
*Leishmania infantum* ^a^	MHOM/CN/94/KXG-Liu	Karamay, Xinjiang, China	Homo	L_Liu
*Leishmania donovani* ^b^	MHOM/CN/2016/SCHCZ	Sichuan, China	Homo	L_HCZ

^a^ Identification of the species based on different techniques and genetic markers is controversial. ^b^ The preliminary identification of this isolate belongs to *Leishmania donovani*.

**Table 2 animals-12-00321-t002:** Whole-genome sequence information of *Leishmania* strains downloaded from NCBI.

GenBank Assembly Accession	*Leishmania* Species	Strain	Country	Host	Date
GCA_003730215.1	*Leishmania donovani*	FDAARGOS_361	Sudan	Homo	2018/11/15
GCA_000227135.2	*Leishmania donovani*	BPK282A1	Nepal	Homo	2011/12/13
GCA_000470725.1	*Leishmania donovani*	BHU 1220	India	Homo	2013/09/30
GCA_003719575.1	*Leishmania donovani*	*Ld*CL	Canada	Homo	2018/11/09
GCA_001989955.1	*Leishmania donovani*	MHOM/IN/1983/AG83	India	Homo	2017/02/07
GCA_001989975.1	*Leishmania donovani*	MHOM/IN/1983/AG83	India	Homo	2017/02/07
GCA_000002875.2	*Leishmania infantum*	JPCM5 (MCAN/ES/98/LLM-877)	Spain	Canis	2011/11/08
GCA_900500625.1	*Leishmania infantum*	JPCM5	Spain	Canis	2018/08/08
GCA_003671315.1	*Leishmania infantum*	HUUFS14	Unknown	Homo	2018/10/22
GCA_003020905.1	*Leishmania infantum*	TR01	Turkey	Homo	2018/03/27
GCA_002243465.1	*Leishmania donovani*	pasteur	France	Homo	2017/08/08
GCA_003730175.1	*Leishmania donovani*	FDAARGOS_360	Sudan	Homo	2018/11/15
GCA_000002725.2	*Leishmania major*	Friedlin (MHOM/IL/81/Friedlin)	Israel	Homo	2011/02/14
GCA_000250755.2	*Leishmania major*	SD 75.1	Unknown	Unknown	2012/03/07
GCA_000331345.1	*Leishmania major*	LV39c5	Unknown	Unknown	2013/01/11
GCA_000441995.1	*Leishmania turanica*	LEM423 (MMEL/SU/79/MEL)	Soviet union	Meles	2013/07/29
GCA_000443025.1	*Leishmania gerbilli*	LEM452 (MRHO/CN/60/GERBILLI)	China	Rhombomys	2013/08/25
GCA_000410695.2	*Leishmania arabica*	LEM1108 (MPSA/SA/83/JISH220)	Saudi Arabia	Psammomys	2016/09/16
GCA_000410715.1	*Leishmania tropica*	L590 (MHOM/IL/1990/P283)	Israel	Homo	2013/06/12
GCA_003352575.1	*Leishmania tropica*	MHOM/LB/2015/IK	Lebanon	Homo	2018/08/01
GCA_003067545.1	*Leishmania tropica*	MHOM/LB/2017/IK	Lebanon	Homo	2018/04/23
GCA_000444285.2	*Leishmania aethiopica*	L147 (MHOM/ET/1972/L100)	Ethiopia	Homo	2016/09/30
GCA_003992445.1	*Leishmania aethiopica*	209-622	Ethiopia	Homo	2019/01/04
GCA_000438535.1	*Leishmania amazonensis*	MHOM/BR/71973/M2269	Brazil	Homo	2013/07/18
GCA_005317125.1	*Leishmania amazonensis*	UA301	Colombia	Homo	2019/05/14
GCA_003992505.1	*Leishmania amazonensis*	210-660	French Guiana	Homo	2019/01/04
GCA_000234665.4	*Leishmania mexicana*	MHOM/GT/2001/U1103	Guatemala	Homo	2011/11/08
GCA_003992435.1	*Leishmania mexicana*	215-49	USA	Homo	2019/01/04

**Table 3 animals-12-00321-t003:** Mapping results of the five *Leishmania* isolates.

Sample	Map Reads	Map Rate (%)	Average Depth (%)	Coverage 1× (%)	Coverage 4× (%)	Coverage 10× (%)
L_DD8	10,781,472	93.23	38.46	96.15	95.72	94.66
L_801	9,059,366	93.50	32.53	96.10	95.51	93.70
L_9044	11,278,920	90.41	38.05	96.09	95.68	94.79
L_Liu	11,870,398	93.93	42.81	96.09	95.67	94.79
L_HCZ	7,844,882	90.01	30.12	96.00	95.32	90.54

Map rate: The number of reads matched to the reference genome divided by the number of reads for clean data. Average depth: average sequencing depth. The number of bases aligned to the reference genome divided by the length of the genome. Coverage 1×: the percentage of the genome covered by reads over 1×. Coverage 4×: the percentage of the genome covered by reads over 4×. Coverage 10×: the percentage of the genome covered by reads over 10×.

**Table 4 animals-12-00321-t004:** Exonic mutation in five *Leishmania* isolates.

Sample	Exonic Mutation	Aneuploidy Mutation	Aneuploidy Mutation Rate (%)
L. DD8	883	773	87.54
L. 801	1022	918	89.28
L. 9044	37,796	1648	4.36
L. Liu	38,078	2054	5.39
L. HCZ	49,352	1322	2.68

**Table 5 animals-12-00321-t005:** Mutation analysis of 18 genes in five *Leishmania* isolates.

Mutated Gene	Number of Mutations				
	L_DD8	L_801	L_9044	L_Liu	L_HCZ
protein kinase	5	4	971	980	1196
dynein heavy chain	1	6	353	355	416
ATP-binding cassette protein subfamily A	60	54	110	112	344
kinesin	6	6	226	231	312
uncharacterized protein	69	83	352	345	308
calpain-like cysteine peptidase	0	14	205	193	173
amastin-like surface protein	69	67	132	125	136
ubiquitin hydrolase	0	0	111	115	132
phosphoglycan beta 1	11	9	94	92	100
serine/threonine-protein kinase	1	1	99	100	96
N-acyl-L-amino acid amidohydrolase	0	0	7	7	87
RNA binding protein	0	2	51	51	84
folate/biopterin transporter	13	11	80	90	70
phosphatidylinositol 3-kinase-like protein	0	1	40	40	69
protein-like multidrug resistant	0	0	16	16	65
amastin-like protein	7	4	43	39	56
kinetoplast-associated protein-like protein	0	0	57	56	55
receptor-type adenylate cyclase a	26	29	30	35	53

Genes are ranked according to the number of mutations in the L_HCZ isolate.

## Data Availability

The datasets supporting the conclusions of this article are included within the article. Genome sequencing data sets have been submitted to the National Center for Biotechnology Information (NCBI) sequence reads archive (SRA), which can be accessed under accession number PRJNA600762 (L_DD8, L_9044, L_HCZ, L_Liu and L_801). The datasets release on 30 December 2021 (L_DD8, L_9044 and L_HCZ) and on 28 February 2022 (L_Liu and L_801)).

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
