# Peer review of "Mutation Characteristics and Phylogenetic Analysis of Five Leishmania Clinical Isolates"

_animals, 2022, doi:10.3390/ani12030321_

Round 1

Reviewer 1 Report

In this manuscript the authors performed whole-genome resequencing and phylogenetic analysis on five Leishmania isolates and looked for mutations in the genome that are potentially related to antimony resistance and virulence, especially in the Leishmania HCZ isolate. Even though their results seem to be interesting, there are not any significant advance in the current knowledge of the field with respect the recently published work by Zheng, Chen et al. 2020). In this previously reported manuscript, the authors already described antimony-resistant testing, genomic sequencing and proteomic profiling of a clinical isolate (HCZ) and two drug-susceptible (DD8 and 9044) strains of L. donovani. They performed comparative studies to identify genetic variations and specificities at DNA and protein levels between the resistant and susceptible phenotypes. They reported molecular mechanisms underlying virulence and antimonial resistance in L. donovani by comparing the phenotypic features, genomic variations, and proteomic differences among HCZ isolate, DD8 and 9044 strains.

In conclusions the authors state that L_HCZ has strong virulence and strong drug resistance although it is not described anywhere in the study and was already published. Also, the authors mentioned the use of animals in the study (line 407) even though it is not employed anywhere along the manuscript.

Specific comments/suggestions:

In general, the English must be revised in depth. There are format aspects as “Leishmania” in italics that has to be corrected along the text.

Some Figure Legends are wrong (e.g, Fig 4A).

Some Acronyms (CV) needs to be described.

Author Response

Response to Reviewer 1 Comments

In this manuscript the authors performed whole-genome resequencing and phylogenetic analysis on five Leishmania isolates and looked for mutations in the genome that are potentially related to antimony resistance and virulence, especially in the Leishmania HCZ isolate. Even though their results seem to be interesting, there are not any significant advance in the current knowledge of the field with respect the recently published work by Zheng, Chen et al. 2020). In this previously reported manuscript, the authors already described antimony-resistant testing, genomic sequencing and proteomic profiling of a clinical isolate (HCZ) and two drug-susceptible (DD8 and 9044) strains of L. donovani. They performed comparative studies to identify genetic variations and specificities at DNA and protein levels between the resistant and susceptible phenotypes. They reported molecular mechanisms underlying virulence and antimonial resistance in L. donovani by comparing the phenotypic features, genomic variations, and proteomic differences among HCZ isolate, DD8 and 9044 strains.

Response: Thank the reviewer for these comments on our manuscript. We would like to take this opportunity to express our sincere thanks to the Reviewer 1 who identified areas of our manuscript that needed corrections or modification.

Point 1: the authors mentioned the use of animals in the study (line 407) even though it is not employed anywhere along the manuscript.

Response 1: We are very sorry for the misunderstanding. There were no animal experiments in this manuscript. The use of animals in this manuscript was a clerical error.

Point 2: In general, the English must be revised in depth. There are format aspects as “Leishmania” in italics that had to be corrected along the text.

Response 2: Thanks a lot. We have checked the full text and modified the non-standardization of written of the manuscript.

Point 3: Some Figure Legends are wrong (e.g, Fig 4A).

Response 3: Thanks. Addressed.

Point 4: Some Acronyms (CV) needs to be described.

Response 4: We are very sorry for the misunderstanding. CV should be copy number variation (CNV) and this was a clerical error. The corresponding part of the manuscript has been revised.

* In the manuscript, the revised sections are highlighted in yellow.

Reviewer 2 Report

In this article, the authors performed a thorough analysis for five Leishmania isolates regarding their genomic and phylogenetic background. As all isolates belonged to  the Leishmania donovani complex, the authors used as reference genome the GCF_000227135.1 of the strain Leishmania donovani  BPK282A1. By comparing the whole-genome sequencing and whole-genome phylogenetic analysis of the isolates with known databases, three isolates (L_801, 18 L_9044 and L_Liu) were identified as L. donovani, while one  (L_HCZ) was identified as L. infantum. L_HCZ was also shown to possess high number of mutations of 20 genes related to antimony resistance and virulence highlighting this isolate as a highly virulent and with strong drug resistance strain which must be taken into consideration by the scientific community. The article is well presented and easy to comprehend, the results are clearly prestentes and the conclusions are based on the analysis of the results. I believe that this study will be very a interested reading for the readers of Animals.

Minor corrections:

Line 12: seperate the two words "by the"

Throughout the text and the bibliography, please use italics for Leishmania (e.g., lines: 12, 14, 21, 27, 82, 94, 128, 148, 240, 265, 278, 325)

Line 48: Add the word "is" in the sentence: "...which is trasmitted..."

Line 49: Failure to treat the infection in time will lead to the death of patients.

Line 50: Leishmania spp. that cause VL is mainly  derived from Leishmania donovani complex....

Line 96: "...for 10min..."

Line 278: "...is shown..." instead of  "...was shown...."

Line 299: "...acid phosphatase..."

Line 325: "...are shown.." instead of "...were shown..."

Author Response

Response to Reviewer 2 Comments

In this article, the authors performed a thorough analysis for five Leishmania isolates regarding their genomic and phylogenetic background. As all isolates belonged to the Leishmania donovani complex, the authors used as reference genome the GCF_000227135.1 of the strain Leishmania donovani  BPK282A1. By comparing the whole-genome sequencing and whole-genome phylogenetic analysis of the isolates with known databases, three isolates (L_801, 18 L_9044 and L_Liu) were identified as L. donovani, while one (L_HCZ) was identified as L. infantum. L_HCZ was also shown to possess high number of mutations of 20 genes related to antimony resistance and virulence highlighting this isolate as a highly virulent and with strong drug resistance strain which must be taken into consideration by the scientific community. The article is well presented and easy to comprehend, the results are clearly presents and the conclusions are based on the analysis of the results. I believe that this study will be very a interested reading for the readers of Animals.

Response: Thank the reviewer for these comments on our manuscript. We would like to take this opportunity to express our sincere thanks to the Reviewer 2 who identified areas of our manuscript that needed corrections or modification.

Point 1: Line 12: seperate the two words "by the"

Response 1: Thanks. Addressed.

Point 2: Throughout the text and the bibliography, please use italics for Leishmania (e.g., lines: 12, 14, 21, 27, 82, 94, 128, 148, 240, 265, 288, 363)

Response 2: Thanks. Addressed.

Point 3: Line 48: Add the word "is" in the sentence: "...which is trasmitted..."

Response 3: Thanks. Addressed.

Point 4: Line 49: Failure to treat the infection in time will lead to the death of patients.

Response 4: Thanks. Addressed.

Point 5: Line 50: Leishmania spp. that cause VL is mainly derived from Leishmania donovani complex....

Response 5: Thanks. Addressed.

Point 6: Line 96: "...for 10min..."

Response 6: Thanks. Addressed.

Point 7: Line 278: "...is shown..." instead of "...was shown...."

Response 7: Thanks. Addressed.

Point 8: Line 299: "...acid phosphatase..."

Response 8: Thanks. Addressed.

Point 9: Line 325: "...are shown..." instead of "...were shown..."

Response 9: Thanks. Addressed.

* In the manuscript, the revised sections are highlighted in yellow.

Reviewer 3 Report

The manuscript “Whole-genome resequencing, mutation characteristics and phy- 2 logenetic analysis of five Leishmania clinical isolates”  uses Genome-wide mutation analysis to identify mutations related to drug resistance and virulence of the newly isolated L_HCZ. Compared with the other four Leishmania isolates, L_HCZ had the most mutations in genes associated with antimony resistance, including the ABC transporter, ascorbate-dependent peroxidase, gamme-glutamylcysteine synthetase, glucose-6-phosphate 1-dehydrogenase, ATP-binding cassette protein subfamily A and multi-drug resistance protein-like genes. The paper is generally well written, the topic is very interesting and well developed.

Some modifications are required:

- Line 12: add space in “bythe”

- Table 1 change “Organism name” to “ Leishmania species”

-Line 87- change “ Culture of Leishmania” in “ “Leishmania culture”

- Table 2 change “Organism name” to “ Leishmania species”

Finally,  I recommend that the paper should be accepted for the publication in this journal.

Author Response

Response to Reviewer 3 Comments

The manuscript “Whole-genome resequencing, mutation characteristics and phylogenetic analysis of five Leishmania clinical isolates” uses Genome-wide mutation analysis to identify mutations related to drug resistance and virulence of the newly isolated L_HCZ. Compared with the other four Leishmania isolates, L_HCZ had the most mutations in genes associated with antimony resistance, including the ABC transporter, ascorbate-dependent peroxidase, gamme-glutamylcysteine synthetase, glucose-6-phosphate 1-dehydrogenase, ATP-binding cassette protein subfamily A and multi-drug resistance protein-like genes. The paper is generally well written, the topic is very interesting and well developed.

Response: Thank the reviewer for these comments on our manuscript. We would like to take this opportunity to express our sincere thanks to the Reviewer 3 who identified areas of our manuscript that needed corrections or modification.

Point 1: Line 12: add space in “by the”

Response 1: Thanks. Addressed.

Point 2: Table 1 change “Organism name” to “Leishmania species”

Response 2: Thanks. Addressed.

Point 3: Line 87- change “Culture of Leishmania” in “Leishmania culture”

Response 3: Thanks. Addressed.

Point 4: Line 49: Table 2 change “Organism name” to “Leishmania species”

Response 4: Thanks. Addressed.

* In the manuscript, the revised sections are highlighted in yellow.

Reviewer 4 Report

The work of Zheng & He reports genome analysis of 5 new Leishmania sp. isolates from India & China. In my opinion, the work brings new knowledge and, hence, is publishable upon mandatory revision. 

1) Introduction must be extended.  The following references (Leishmania biology, taxonomy, genomics) should be added and discussed in the context:

10.1371/journal.pntd.0004349

10.1098/rsob.200407

10.3390/pathogens10091124

10.1016/j.pt.2014.12.012

Also add a few sentences about Trypanosomatidae in general: 10.1017/S0031182018000951  

2) Please make sure to Italicize species and genera. This is not the case in many instances. 

3) Please add ALL the necessary references to the Materials and Methods section. Every program should be properly cited.

4) Please provide accession numbers for the newly sequenced genomes. They musty be opened. 

5) Figures 1, 2 and 3 must be presented as columns (lines make no sense). 

6) I suggest to replace Fig. 5 by the Table.

7) Fig. 5 and 6 must have proper legends. 

8) Please provide statistical supports for the tree: a) it must be re-inferred using ML and Bayesian algorithms. b) Methods must be properly described. In the current version it is not clear how it was built.

9) It may also be interesting to look at the recombination or gene conversion loci identified in some recent publications, such as this L. donovani-focused one: 10.3390/pathogens9070572 although I would understand if this would go beyond the scope of this current work. 

Author Response

Response to Reviewer 4 Comments

The work of Zheng & He reports genome analysis of 5 new Leishmania sp. isolates from India & China. In my opinion, the work brings new knowledge and, hence, is publishable upon mandatory revision. 

Response: Thank the reviewer for these comments on our manuscript. We would like to take this opportunity to express our sincere thanks to the Reviewer 4 who identified areas of our manuscript that needed corrections or modification.

Point 1: Introduction must be extended. The following references (Leishmania biology, taxonomy, genomics) should be added and discussed in the context:

10.1371/journal.pntd.0004349

10.1098/rsob.200407

10.3390/pathogens10091124

10.1016/j.pt.2014.12.012

Also add a few sentences about Trypanosomatidae in general: 10.1017/S0031182018000951 

Response 1: Thanks. Relevant contents and references have been added to the introduction of the manuscript. For details, please see the cf. [5-11] and cf. [18]

Point 2: Please make sure to Italicize species and genera. This is not the case in many instances.

Response 2: Thanks. Addressed.

Point 3: Please add ALL the necessary references to the Materials and Methods section. Every program should be properly cited.

Response 3: Thanks. All programs were cited

Point 4: Please provide accession numbers for the newly sequenced genomes. They must be opened.

Response 4: Thanks. This part has been added to the Data Availability Statement of the manuscript.

Point 5: Figures 1, 2 and 3 must be presented as columns (lines make no sense).

Response 5: Thanks. The figures 1, 2 and 3 have been modified and presented as columns.

Point 6: I suggest to replace Fig. 5 by the Table.

Response 6: Thanks a lot for your suggestion. However, it would be too tedious if we replaced Figure 5 to a table, the comparison matrix of table was more complicated than a Fig. In addition, we made the Fig. 5 according to a complicated table. Therefore, we decided to use the form of figure to show the results more reasonable and concise.

Point 7: Fig. 5 and 6 must have proper legends

Response 7: Thanks. More information was provided in the legends

Point 8: Please provide statistical supports for the tree: a) it must be re-inferred using ML and Bayesian algorithms. b) Methods must be properly described. In the current version it is not clear how it was built.

Response 8: a) the three was rebuilt using the ML algorithm by iqtree with a bootstrap replicate of 1000. b) the methods were added in the material and methods section (Line xx-xx).

Point 9: It may also be interesting to look at the recombination or gene conversion loci identified in some recent publications, such as this L. donovani-focused one: 10.3390/pathogens9070572 although I would understand if this would go beyond the scope of this current work.

Response 9: Thanks for your comments and suggestions. Gene recombination or gene conversion loci identified in leishamania was very meaningful, which require a lot of time and efforts at the present stage for us. We will focus on these aspects in the future research.

* In the manuscript, the revised sections are highlighted in yellow.

Round 2

Reviewer 1 Report

Even though their manuscript has been improved with the corrections, I still considering that there is not any significant advance in the current knowledge of the field with respect the recently published work by Zheng, Chen et al. 2020).

Author Response

Point 1: Even though their manuscript has been improved with the corrections, I still considering that there is not any significant advance in the current knowledge of the field with respect the recently published work by Zheng, Chen et al. 2020).

Response 1: We appreciate the reviewer for the critical comments. However, we would consider that the previous paper (Zheng et al., Parasites & Vectors, 2020) focused on verifying the virulence and antimony-resistance of the Leishmania clinical isolate (L_HCZ) at molecular levels, using an integrative approach of genome sequencing, proteomic profiling and phenotypic analysis. By contrast, the current manuscript is to investigate/understand the genetic evolution of this isolate, comparing with other/reference isolates, by exploiting genome sequences publicly available. In this work, together with the re-sequenced genomes, we included genomes of 33 Leishmania protozoa strains for phylogenetic analysis, which clearly would be of help to improve the molecular identification of Leishmania species and our understanding of the potential mechanisms underlying resistance in this important parasite. What’s more, whole-genome data can effectively eliminate the influence of factors such as horizontal gene transfer and differences in gene evolution rate between groups on the phylogenetic tree and have a broad application prospect in phylogenetic analysis and strain identification.

We have slightly modified the title to make this manuscript distinguishable from our previous publications.

* In the manuscript, the revised sections are highlighted in yellow.

Reviewer 4 Report

I liked the revised version  - it is a significant improvement. Some final touches may still be needed.

1) Please cite and discuss this relevant paper: doi:10.3390/pathogens10091124

2) Please replace word protozoa (and all its derivatives) by protist (and derivatives, respectively)

3) ln. 371 Italic for Leishmania

4) Please cite actual papers in the methods - links to the web-sites are not sufficient.  Most (if not, all) of these programs are published in actual Journals. 

5) Please make sure that all the space on pages is used up. Half-empty pages are rather annoying. 

Author Response

Point 1:  Please cite and discuss this relevant paper: doi:10.3390/pathogens10091124

Response 1: Thanks. The reference has been added to Introduction of the manuscript. For details, please see the cf. [11] (L67-68).

Point 2: Please replace word protozoa (and all its derivatives) by protist (and derivatives, respectively)

Response 2: Thanks. Addressed (L59).

Point 3: ln. 371 Italic for Leishmania

Response 3: Thanks. Addressed (L326).

Point 4: Please cite actual papers in the methods - links to the web-sites are not sufficient. Most (if not, all) of these programs are published in actual Journals. 

Response 4: Thanks. Relevant references have been added to Materials and Methods of the manuscript. For details, please see the cf. [22-31] (L155-175).

Point 5: Please make sure that all the space on pages is used up. Half-empty pages are rather annoying. 

Response 5: Thanks. Addressed.

* In the manuscript, the revised sections are highlighted in yellow.
